

# Pairing algorithm for varying data in cluster based heterogeneous wireless sensor networks

Zahida Shaheen[1,*], Kashif Sattar[2,*] and Mukhtar Ahmed[3,4]

[1] Pir Mehr Ali Shah Arid Agriculture, University Institute of Information Technology (UIIT), Rawalpindi, Punjab, Pakistan
[2] Pir Mehr Ali Shah Arid Agriculture University, University Institute of Information Technology (UIIT), Rawalpindi, Punjab, Pakistan
[3] Pir Mehr Ali Shah Arid Agriculture University, Department of Agronomy, Rawalpindi, Punjab, Pakistan
[4] Department of Biological Systems Engineering, Washington State University, Pullman, United States of America
[*] These authors contributed equally to this work.

Corresponding author
Mukhtar Ahmed,
ahmadmukhtar@uaar.edu.pk

## ABSTRACT

In wireless sensor networks (WSNs), clustering is employed to extend the network's lifespan. Each cluster has a designated cluster head. Pairing is another technique used within clustering to enhance network longevity. In this technique, nodes are grouped into pairs, with one node in an active state and the other in a sleep state to conserve energy. However, this pairing can lead to communication issues with the cluster head, as nodes in sleep mode cannot transmit data, potentially causing data loss. To address this issue, this study introduces an innovative approach called the "Awake Sleep Heterogeneous Nodes' Pairing" (ASHNP) algorithm. This algorithm aims to improve transmission efficiency in WSNs operating in heterogeneous environments. In contrast, Energy Efficient Sleep Awake Aware (EESAA) algorithm are customized for homogeneous environments (EESAA), while suitable for homogeneous settings, encounters challenges in handling data loss from sleep nodes. On the other hand, Energy and Traffic Aware Sleep Awake (ETASA) struggles with listening problems, limiting its efficiency in diverse environments. Through comprehensive comparative analysis, ASHNP demonstrates higher performance in data transmission efficiency, overcoming the shortcomings of EESAA and ETASA. Additionally, comparisons across various parameters, including energy consumption and the number of dead nodes, highlight ASHNP's effectiveness in enhancing network reliability and resource utilization. These findings underscore the significance of ASHNP as a promising solution for optimizing data transmission in WSNs, particularly in heterogeneous environments. The analysis discloses that ASHNP reliably outperforms EESAA in maintaining node energy, with differences ranging from 1.5% to 10% across various rounds. Specifically, ASHNP achieves a data transmission rate 5.23% higher than EESAA and 21.73% higher than ETASA. These findings underscore the strength of ASHNP in sustaining node activity levels, showcasing its superiority in preserving network integrity and ensuring efficient data transmission across multiple rounds.

# INTRODUCTION

To enhance network stability, the lifetime of the network throughout the heterogeneity process is used expansively. Algorithms of routing are proposed based on heterogeneity that works on processes of clustering, the lifetime of a network, stability and efficient energy coming forward (*Chaurasia, Kumar & Kumar, 2023*; *Wu et al., 2022*; *Hao, Hong & He, 2022*; *Yang et al., 2022*; *Liu, Liu & Wang, 2023*). Heterogeneous wireless sensor networks (WSNs) are made out of sensor nodes that have changed abilities, remembering contrasts for processing power, detecting range, energy assets, and correspondence range. This variety in nodes qualities empowers the network to screen and gather information across various conditions and applications effectively. By utilizing the qualities of each type of nodes, heterogeneous WSNs can accomplish further developed inclusion, adaptability, and energy proficiency in contrast with homogeneous networks where all nodes have indistinguishable abilities. Wireless sensor networks are an emerging technology for monitoring physical world. The energy constraint of wireless sensor networks makes energy saving and prolonging the network lifetime become the most important goals of various routing protocols. Clustering is a key technique used to extend the lifetime of a sensor network by reducing energy consumption. Also, putting few heterogeneous nodes in wireless sensor network is an effective way to increase the network lifetime and stability (*Liu, Liu & Wang, 2023*).

In a heterogeneous WSN, the detecting capacities of sensor devices can shift fundamentally dependent upon their plan and expected application. Here are a few vital parts of detecting capacities that can contrast among sensor nodes for instance in detecting range, the sensor nodes might have various reaches over which they can distinguish uniqueness or occasions in detecting modalities, Sensor nodes might be furnished with various kinds of sensors to recognize different environmental parameters. In detecting accuracy, the precision of sensor estimations might fluctuate among nodes. The rate at which sensor nodes test and gather information can shift is called sampling rate. Sensor nodes with various detecting capacities might consume differing measures of power during detecting activities in the class of power utilization. Depending upon their computing power and locally available handling abilities, sensor nodes might perform various degrees of information handling locally prior to sending information to the base station or different nodes in the networks. Nodes with higher computing power can perform more complex information handling errands, for example, signal processing or data gathering is the significant part of data process.

The pairing concepts are familiarized to improve the lifetime and the stability of a network for this purpose. Similar process is applied for sensors nodes located at the lowest distance between for pairing to ensure loss less communications. The nodes switch between "Sleep" and "Awake" mode. Nodes in Sleep-mode save the energy whereas the nodes in active-mode gather data from surroundings and transmit this data to CHs as shown in Fig. 1.

The hybrid approach, proposed by *Shagari et al. (2020b)*, is known as Energy and Traffic Aware Sleep and Awake (ETASA). There are four processes in this approach. These processes

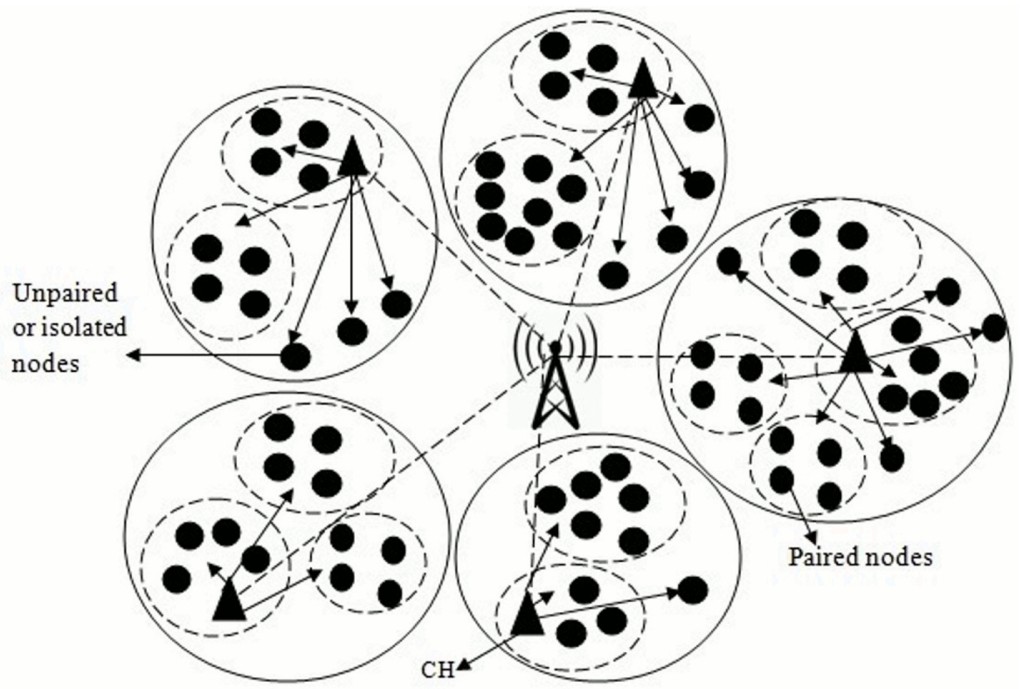

**Figure 1** Paired node and isolated node.

which are to be done in chunks are the selection of CH after pairs formation, scheduling of TDMA, and traffic and energy approach of sleep-awake awareness. When we go through the initial pair formulating process, there is an introduction to the pairing technique. In this technique, the sensor nodes having similar applications are to be positioned at smaller distances from one another and within similar communication intra cluster range. This is due to the element that the nodes of a sensor as positioned in capturing close proximity of the similar data and forwarded to the base station. Therefore, outcomes are delivered as the transmission of data that is not needed and collision that damages the performance of the network. The nodes are paired as substitutes for data sensing and transmission. Then the mechanism of sleep-awake for traffic and energy awareness is familiarized between the pairs. Consequently, within the nodes of the pair only a single node can perform data sensing at the time for transmission of tasks to CH. The other pairs are kept in the sleep mode by concluding communication to preserve the energy with the CH. The centralized approach is used for pairs formation. During the initial deployment of the network, every sensor node is reconfigured with location. The position is adjusted by each node. Every node passes its adjustment and ID information towards the BS. This information is utilized by BS for the calculation of distances within the member nodes. Functionality, which is the same and in accordance with the application, is paired with the nodes that are less than ten meters away in the intra-cluster communication range. So, all the information that is paired is broadcast to BS by all network nodes. At this stage, each node of the member is well-known to its pairs. Therefore, the nodes that are without pairs remain in isolation

since they are not close to other nodes. There are efforts currently in process to perform efficient routing.

The solution to this problem is retaining duty-cycle in routing of the cluster in order to reduce redundant transmission. Sleep Awake Energy Efficient Distributed (SEED) is one of the state-of-art routing schemes that uses duty-cycling in order to reduce redundant transmissions towards the sensor nodes that are in proximity to achieving efficient energy.

Therefore, routing schemes other than SEED suffer from the problem of idle listening, which is directed to unnecessary consumption of energy throughout the network. Idle listening is the time duration when sensor nodes are used for trans-receiving of active radio, but no data is received or transmitted by the sensor node (*Daneshfar & Maihami, 2014*). SEED uses Time Division Multiple Access (TDMA) scheduling between the members of the clusters as CMS which is directed to idle listening issues in network (*Daneshfar & Maihami, 2014*). Using old conventional TDMA, each member of cluster is allocated one slot. Moreover, the sensor nodes should be awakened and transfer its radio to "ON" during the process of scheduling, if no data is sent towards the CH. So, being in an idle state the nodes are operated when the sufficient energy level is consumed.

Therefore, CH operates in the idle state which wastes energy and the idle slots of time in current state. Furthermore, SEED cannot survive with the sensor nodes trans-receiving heterogeneous data. In nodes' pairing, the alternation is done in a Round Robin manner into awaken and sleep mode without taking into consideration the heterogeneity among the sensor nodes. Therefore, the sensor nodes with more traffic rate disperses more energy in contrast to the sensor nodes with low transmission rate (*Sharma & Bhondekar, 2018*).

Furthermore, in SEED, the new mode might not have enough energy for transmission of data at a given speed. In SEED, clustering is done by the sleep awake mechanism (*Ahmed et al., 2016*). In this process, dual or more nodes that are within the similar transmission of similar application range to one another, are grouped together for the formation of sub-clusters. In a paired grouping of sub-cluster, only the single sensor node gets awakened, senses the surroundings, and then directs its sensed data to the CH while the leftover nodes stay in the sleep mode to reserve power consumption of battery. But there is a drawback to this approach as there is no consideration of energy within the sub-cluster of nodes during the alternation of awake and sleep modes. SEED did not consider heterogeneity of traffic between the sensor nodes for alternations of awake and sleep, and that does not make it considerable with sensor nodes of heterogeneous traffic for WSNs. So, in TDMA, as the conventional SEED, the consumption of energy is enhanced. Moreover, the energy effect of heterogeneity has been established for many years. In the previous years, the mode of communication has changed rapidly. Communication can be done without physical existence and is known as wireless communication. The modern style of wireless technologies has developed so rapidly and is getting more fame due to low cost and its complexity. Moreover, the network established where in wireless communication happens between the sensors is known as a WSN (*Singh & Gupta, 2016*). The sensor nodes that are deployed to communicate with one another, sense the environment, and then finally transmit the data towards the destination, which is a BS (base or sink station). To formulate an explicit task, the nodes are grouped together for transmission of data, and these grouped

nodes form a cluster (*Goyal, 2014*). Clustering is helpful in improving efficiency of the defined network. Recently, the nodes in the cluster are selected based on the equation of the probability that is in consideration for data transmission towards the BS. In the Centralized Energy Efficient Clustering (CEEC) protocol, the heterogeneity concept is introduced by the authors (*Aslam et al., 2012*). The heterogeneity of three levels has been introduced by the authors. The area of network is divided into three rectangular sections that are equal in size. They are named the region of high energy that has super nodes alone, region of medium energy that has advanced nodes and the region of low energy that has normal nodes only. Similar type nodes can be grouped in clusters. The selection of a CH is the responsibility of a BS in every phase. A CH is selected on the basis of the distance and the energy. To send the data to the CHs, the energy is lost rapidly, which leads to network stability and lifetime being affected. CEEC selects an even number of CHs for each round and ensures stability in the network. A heterogeneous protocol, known as Balanced Energy Efficient Network Integrated Super Heterogeneous Protocol (BEENISH) (*Qureshi et al., 0000*). In BEENISH, there are four heterogeneous levels of nodes, known as ultra-super, super, advanced, and normal. These nodes are normally spread all over the network and choice of a CH is made based purely on leftover energy and average energy for deployment of nodes. In BEENISH, nodes with maximum energy form a CH. Selection of CHs with high energy have more lifetime and stability. The authors suggested an approach known as Distributed Energy Efficient Clustering (DEEC) (*Singh & Gupta, 2016*). In DEEC, the selection of a CH is done by probability based on the leftover energy and average in DEEC enhances the lifetime of the network and the network stability. The nodes having high energy form a CH whereas the low energy nodes are allowed to sense, collect, and transmit data towards the CHs.

*Shah, Javaid & Qureshi (2012)* proposed an approach, termed as Energy Efficient Sleep Awake Aware (EESAA) where in an intelligent homogeneous routing protocol is implemented. In EESAA, pairing of the nodes is introduced by the author. In pairing, only one node of pair is in awake state and responsible for sending the data towards the CH in every round. The nodes other than the pair remain in the sleep state of sleep to preserve the energy. The CHs are purely selected based on residual energy of nodes. The awake sleep strategy not only reserves the energy of nodes but also halts the transmission of redundant data. Moreover, in EESAA, such nodes are not paired with further nodes and remain active through the entire process. Energy is lost by these active nodes, which leads to reduction in network stability.

*Meenakshi, Ahmad & Nazeer (2024)* discuss that the cluster head selection process is guided by multiple factors, including residual energy, proximity to neighbors, distance to the base station, node degree, and node centrality. Additionally, an algorithm called Heuristic Wing Ant-fly Optimization (HWAFO) is employed to determine the optimal path between the cluster head and the Base Station (BS) based on factors such as distance, residual energy, and node degree. The study evaluates the system's performance by analyzing the number of active nodes, their energy consumption, and the quantity of data packets received by the BS.

There have been multiple activities to ensure that routing algorithms should work with keeping conserving energy in consideration and equally balancing the load even amongst heterogeneous sensor nodes to sustain the network lifespan. There have been several existing methods that have tried to work in terms of energy efficiency and minimizing imbalanced load like SEED, Traffic and Energy Aware Routing (TEAR), Energy and Traffic Aware Sleep Awake (ETASA), *etc*. These techniques work efficiently. However, they have many drawbacks like idle listening problem, inefficient working in an environment with heterogeneous traffic rate surrounding the sensor nodes, wastage of energy transmitted from the nearby nodes, *etc*. To overcome these issues, we introduced a novel routing algorithm in this work. The new routing algorithm is termed as Awake Sleep Heterogeneous Nodes' Pairing (ASHNP) scheme. It involves the pairing of nodes deployed in WSNs using two thresholds. We made improvement in pairing strategy in which if the data from all nodes in ten meters' vicinity is the same, then pairing strategy implements otherwise nodes with different data can be considered as isolated and awake as shown in Fig. 1. This overcomes the data loss. The simulation results prove the proposed work's efficacy in terms of various metrics like number of dead nodes, throughput, number of packets sent. Moreover, the proposed algorithm is compared with Energy Efficient Sleep Awake Aware (EESAA) to show its feasibility.

An essential knowledge gap exists concerning the loss of important data resulting from this sleep-state of sensor nodes in the current pairing technique (*Pirretti et al., 2006*; *Znaidi, Minier & Ubéda, 2013*; *Shagari NM et al., 2020a*; *Khan et al., 2013*; *Płaczek, 2024*; *Nordin, Ismail & Abu-AlShaeer, 2018*; *Mohammadi & Shirmohammadi, 2023*; *Abdulzahra, Al-Qurabat & Abdulzahra, 2023*; *Kantzavelou et al., 2022*; *Rahman et al., 2021*; *Van Truong, Nayyar & Masud, 2021*; *Sheikhpour, Jabbehdari & Khadem-Zadeh, 2011*). While energy-saving measures are paramount in WSNs, the trade-off between energy efficiency and data loss remains a critical challenge that permits further investigation. Therefore, understanding and mitigating the effects of data loss within the circumstance of pairing techniques represent crucial areas for future research and development in WSN clustering approaches. Summary of existing techniques have been shown in Table 1.

The main idea of EESAA is pairing or grouping the nodes for homogenous network for minimize the energy consumption. In EESAA only two nodes are paired with each other and in these paired nodes the one node is in sleep state for one communication interval which creates the data loss. In WSN's Clustering using pairing technique, only one node among pairs is in the awake-state which senses the data and sends it to the cluster head, whereas other nodes in a pair are in a sleep-state. The sleep-state of pairing nodes gives us the advantage of energy-saving and minimizes computation and communication costs. However, it causes the important data loss in current pairing technique. The main focus of this article is to increase the data accuracy by detecting and transmitting important data missed by sleep nodes.

## CONTRIBUTIONS

The major contributions of the proposed work are given as follows.
- The nodes deployed in the WSN are compared to form pairs.

**Table 1  Summary of existing techniques.**

| Protocol | Energy efficient | Pairing | Heterogeneous | Data accuracy |
|---|---|---|---|---|
| Low-Energy Adaptive Clustering Hierarchy-LEACH (2000) | YES | NO | NO | NO |
| Hybrid Energy-Efficient Distributed clustering-HEED (2004) | YES | NO | NO | NO |
| Energy efficient sleep awake aware-EESAA (2012) | YES | NO | NO | YES |
| Stable Election Protocol-SEP (2012) | YES | NO | YES | YES |
| Developed distributed energy-efficient clustering-DEEC (2014) | YES | NO | YES | NO |
| Sleep-awake aware–SAA (2016) | NO | YES | NO | NO |
| Traffic and energy aware routing protocol-TEAR (2018) | YES | YES | YES | NO |
| Distributed Efficient Fuzzy Logic–DEFL (2016) | YES | NO | YES | NO |
| Energy efficient reservation-based cluster head selection EERCH (2018) | YES | NO | NO | YES |
| Sleep-awake Energy Efficient Distributed-SEED (2020) | YES | YES | YES | NO |
| Traffic Aware Sleep-Awake Cluster-Based (ETASA) (2020) | YES | YES | YES | NO |
| Awake Sleep Heterogeneous Nodes' Pairing (ASHNP) | YES | YES | YES | YES |

●A novel algorithm, termed as Awake Sleep Heterogeneous Nodes' Pairing (ASHNP), is proposed.

●The pairing of nodes is done depending upon the distance between them and their states.

In the context of WSNs energy-efficiency, ASHNP discourses several technical challenges to increase resource utilization and prolong network durability. ASHNP addresses the technical challenges of energy-efficiency in WSNs by optimizing node pairing and path selection, dynamically adapting to changing network conditions, and implementing holistic resource management strategies. These efforts contribute to enhancing network reliability, prolonging node lifespan, and maximizing energy efficiency in WSN deployments.

The remainder of the article is coordinated as follows. Section 'Introduction' presents the overview of the research work. The Section 'System model' included proposed system model and presumptions are illustrated. Section 'Simulation results' talks about the proposed approach and deployed simulation and presentation assessment. Section 'Discussion' contains some discussions and at the last, conclusions of the all presumptions.

## SYSTEM MODEL

The data is lost to a great extent without there being a threshold of time and data. In our proposed scheme, ASHNP, after a specific threshold time, data coming from all nodes is checked. If data difference is less than data threshold values, then the energy level of nodes is computed, and nodes' sleep awake states are updated. Due to time and date thresholds, the data loss is less as compared to previous techniques. Figure 2 shows the pairing of

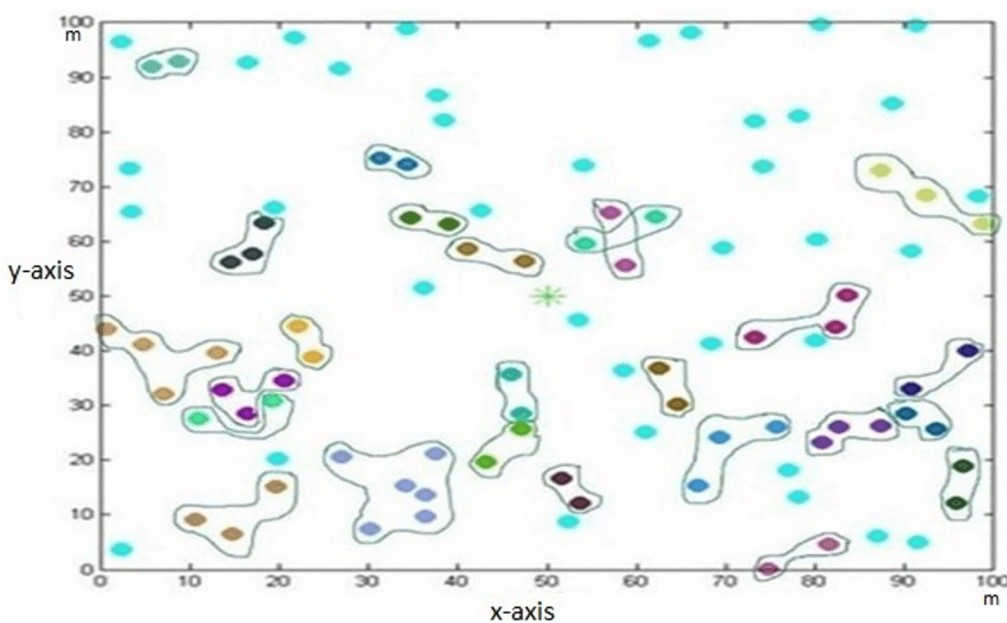

**Figure 2**   Formation of pairs.

nodes. Every pair is shown in different color. In a pair, every node has the same data and nodes are a specific distance apart from each other.

There is a process of nodes' pairing described in Algorithm 1. The algorithm should try to gain load-balancing and efficient energy between the heterogeneous sensor nodes in order to extend the lifetime of the network. A significant number of heterogeneous wireless sensor nodes are installed in the proposed effort. These sensor nodes are built for data collecting and sensing in a variety of environments and include wireless connection capabilities. Some key features of the proposed work are discussed below.

## Heterogeneity

Because the sensor nodes in the WSN are diverse, they have a range of capabilities and qualities (*Fedorenko et al., 2023*; *Gupta & Singh, 2023*; *Singh, Garg & Malik, 2023*). In terms of processing power, energy resources, sensing range, sensing modalities, memory a capacity, and communication capabilities, this heterogeneity might vary. Heterogeneity enables adaptability to various application needs and resource utilization optimization.

## Sensing capabilities

One or more sensors are installed in each sensor node to gather environmental data. The sensors might be any kind of sensor that is appropriate for the application, such as temperature sensors, humidity sensors, light sensors, motion sensors, gas sensors, etc. The nodes may have various sensor configurations, enabling them to record various characteristics of the environment.

## Energy efficiency

Given that WSN nodes are frequently powered by batteries or energy-harvesting devices, energy efficiency is a crucial issue. Energy management strategies are used to optimize power usage since the sensor nodes work under power limits. To increase the network's lifespan, this might use energy harvesting methods, duty cycling, or sleep/wake scheduling.

## Mobility

Some sensor nodes in the network may be mobile or able to move, depending on the application. In terms of routing, data aggregation, and network topology maintenance, mobility poses new issues (*Dey, Bandyopadhyay & Nandi, 2023*; *Shaker et al., 2023*). To manage mobility, techniques like dynamic cluster reformation (*Zhao & Zhang, 2023*) or node tracking (*Sowndeswari & Kavitha, 2023*) algorithms may be used. The above given parameters are the most important while constructing a cluster based heterogeneous WSN. In the proposed network, nodes are paired with each other, which reduces the overall network computational complexity and increases network efficiency. In the network, nodes' status changes after every 2 s, which ensures increased network throughput.

In existing schemes, pairing occurs only one time whereas in our proposed approach the pairing process repeats itself after a certain time threshold as shown in Fig. 3. If data variation is higher than a certain data threshold, then that data are also important and need to transmit to the cluster head. During this process CH may be changed, so CH selection mechanism re-elects new CH based on selecting probability of each candidate node. In the WSN each cluster has a chief, which is called the cluster head and usually perf tasks and aggregation, and several common sensor nodes (SN) as members.

The Cluster Head nodes operate as access points between the sensor nodes and the Base Station. The role of each Cluster Head is to perform common roles and responsibilities for all the nodes in the cluster, like aggregating the data before transmitting it to the Base Station. Ideal cluster head is the one which has the highest residual, the maximum number of neighboring nodes, and the smallest distance from base station. In the concept of energy, pairing, sensor nodes of the same application which are close to each other form a pair (not necessarily two nodes) for data sensing and communication. In a large number of sensors which are densely deployed, the pairing strategy is usually applied. The advantage of pairing is that the nodes in Sleep-mode save their energy by avoiding overhearing and idle listening during Sleep-mode. Also, it minimizes computation and communication overhead. However, the major issue is the data loss of sleep- nodes.

In any building, by installing ASHNP technique, smoke can detect from any sensor node either it is in sleep state. Due to the changed data the defected node converts into awake state and send their information to the CH. Wireless sensor network comprise of spatially sensor nodes that independently gather and send information to a cluster head. This network gives continuous monitoring capacities allowing farmers to resolve problems in view of current field condition.

Proposed pairing methodology is explained in Fig. 4. After the pairing, data is checked from nodes in a pair, after selected threshold time. If data difference is greater than data

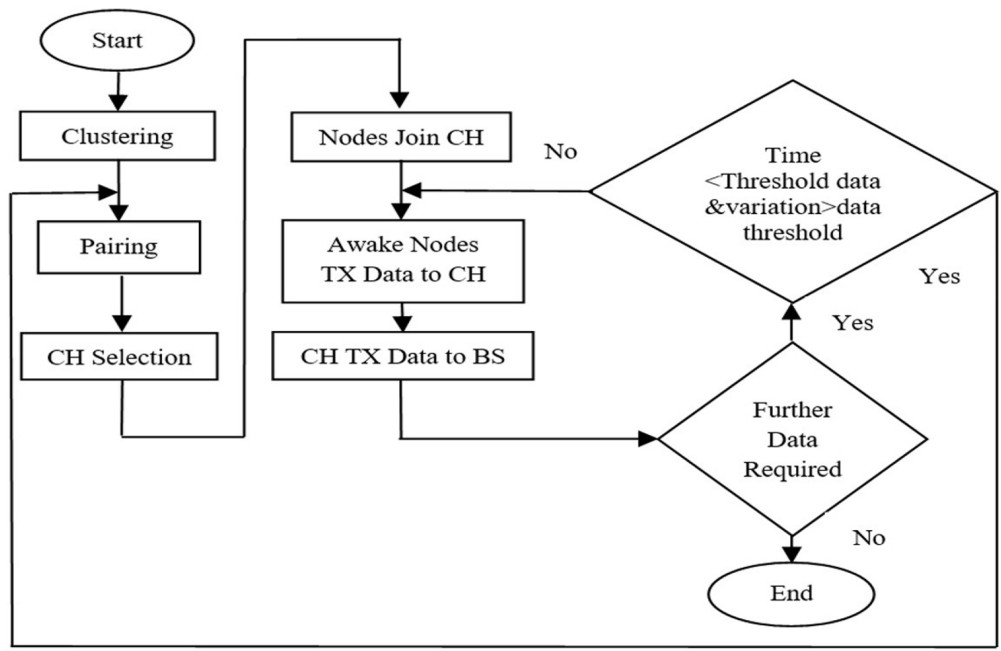

**Figure 3** System model.

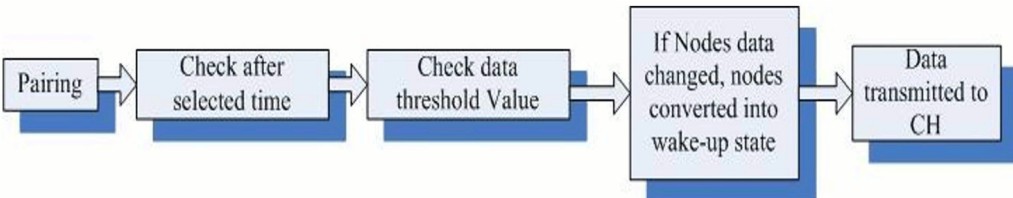

**Figure 4** Proposed pairing mechanism.

threshold value, then nodes are converted into awake-state and transmit its data to CH. These steps are followed in Fig. 4.

There are a few particulars of detecting capacities considered in the ASHNP study:

**Detecting range:** Through numerical analysis, ASHNP assesses what varieties in recognizing range mean for network performance, taking into account factors like coverage area and detection accuracy.

**Detecting modalities:** Sensor nodes might be equipped with various sorts of sensors to identify different environmental parameters. ASHNP conducts a definite analysis of sensor modalities sent inside the network, looking at their viability in capturing relevant data and facilitating in decision-making.

**Detecting accuracy:** Numerical analysis evaluates the effect of identifying accuracy variations on data reliability and network proficiency, featuring the significance and quality control measures.

| | Algorithm 1: Algorithm of pairing |
|---|---|
| 1 | *Initialize* an empty vector to store pairs of node IDs with their statuses (**Pair-Vector**) |
| 2 | *Initialize* an empty set to keep track of paired node IDs (paired-Nodes) |
| 3 | Create a new **pair** and add node *i* to it as an active node |
| 4 | Insert node *i* ID into the set of paired nodes |
| 5 | *Iterate* over other nodes in the cluster to find nodes to pair with node *i* |
| 6 | **for** each node *j* in cluster |
| 7 | Skip nodes that have already been included in pairs |
| 8 | **if** node *j* is in paired-Nodes |
| 9 | continue to the next node *j* |
| 10 | Get the position and temperature of node *j* |
| 11 | Calculate the distance between node *i* and node *j* |
| 12 | Check if nodes have the same temperature and are within **10** meters of each other |
| 13 | **if** node *i* Temperature $==$ node *j* Temperature **AND** distance $<= 10.0$ |
| 14 | **If** conditions are met |
| 15 | Ensure a **unique** node ID for each node across clusters |
| 16 | Create a **pair** with node *i* as active and node *j* as sleep |
| 17 | **pair** $=$ {node *i*, "**Active**"}, {node *j*, "**Sleep**"} |
| 18 | Insert node *j* into the set of paired nodes |
| 19 | **End** of iteration over other nodes |
| 20 | Add the current pair to the list of pairs if there are more than one node in the pair |
| 21 | **if** pair size >1 |
| 22 | add pair to Pair-Vector |
| 23 | **else** |
| 24 | add node *i* to isolated nodes list as an active node |
| 25 | **End** of algorithm |

**Sampling rate:** The rate at which sensor nodes test and gather data can shift fundamentally, influencing data procurement proficiency and power utilization. ASHNP analyses the impacts of various inspecting rates on network performance, taking into account compromises between data granularity and energy consumption.

**Power utilization:** Sensor nodes with diverse distinguishing capacities might show shifting degrees of energy utilization during detecting activities. ASHNP conducts troublesome energy utilization investigation to make sense of the connection between detecting capacities, power usage, and network durability.

**Data processing:** Depending on their computing power and processing capabilities, sensor nodes may perform varying degrees of data processing locally before transmitting information to the base station or other nodes. ASHNP evaluates the efficacy of different data processing strategies in optimizing network bandwidth utilization and minimizing latency.

Through complete numerical analysis of these detecting capacities, the ASHNP study gives important insights into the complex transaction between sensor characteristics and network performance inside heterogeneous WSN environments. These insights illuminate

**Table 2  Simulation parameters and values.**

| Parameter | Value |
|---|---|
| Network area | 100*100 m |
| Total number of nodes | 100 |
| Total number of rounds | 4,000 |
| Initial energy | 0.5 Joule |

the advancement regarding upgraded detecting and data processing techniques, at last improving the efficiency and reliability of WSN deployments.

## SIMULATION RESULTS

In this section, the simulation results are being discussed. The section comprises of simulation setup followed by simulation results' discussion.

### Simulation environment

The simulations in the proposed work are performed using an Intel Core i3 machine with 6 GB RAM, and 500 GB HDD. For performing the simulations, this part communicates the detail depiction of simulation. There are 100 nodes are deployed in a 100m*100m area for monitoring. The base station is located at the center of region as shown in the Fig. 5.

We executed the proposed approach by recreation utilizing NS3 simulator. The NS3 simulator is a discrete-occasion network simulator designated fundamentally for research permitting clients to run genuine execution code in the simulator. Over the span of this development, we followed same parameters for network simulation as species in *Shah, Javaid & Qureshi (2012)* and *Shagari NM et al. (2020c)* contained in Table 2 beneath. Moreover, a total of 100 sensor nodes are being deployed in a 100 by 100 meter-square region. The simulation setup reflects a typical deployment scenario in WSNs, where sensor nodes are strategically positioned to monitor environmental conditions. By observing to established parameters and utilizing advanced simulation tools like NS3, researchers can gain valuable insights into network behavior and performance, ultimately driving innovation and improvement in WSN technologies.

### Simulation parameters

The proposed model is evaluated in terms of different performance parameters. These parameters are of vital importance and are extensively used in the literature (*Shah, Javaid & Qureshi, 2012*; *Qureshi et al., 0000*; *Aslam et al., 2012*; *Goyal, 2014*; *Daneshfar & Maihami, 2014*; *Singh & Gupta, 2016*; *Ahmed et al., 2016*; *Sharma & Bhondekar, 2018*; *Fedorenko et al., 2023*; *Gupta & Singh, 2023*; *Singh, Garg & Malik, 2023*; *Dey, Bandyopadhyay & Nandi, 2023*; *Shaker et al., 2023*; *Zhao & Zhang, 2023*; *Sowndeswari & Kavitha, 2023*; *Guo, Chen & Li, 2023*; *Gao et al., 2023*; *Liu et al., 2023*; *Wen et al., 2023*; *Shagari et al., 2020c*). Some of these parameters are as follows:

**Energy consumption**: In a routing network, energy consumption by nodes refers to the amount of energy that individual nodes within the network require to perform their functions and tasks. Each node in the network, such as routers or switches, typically has

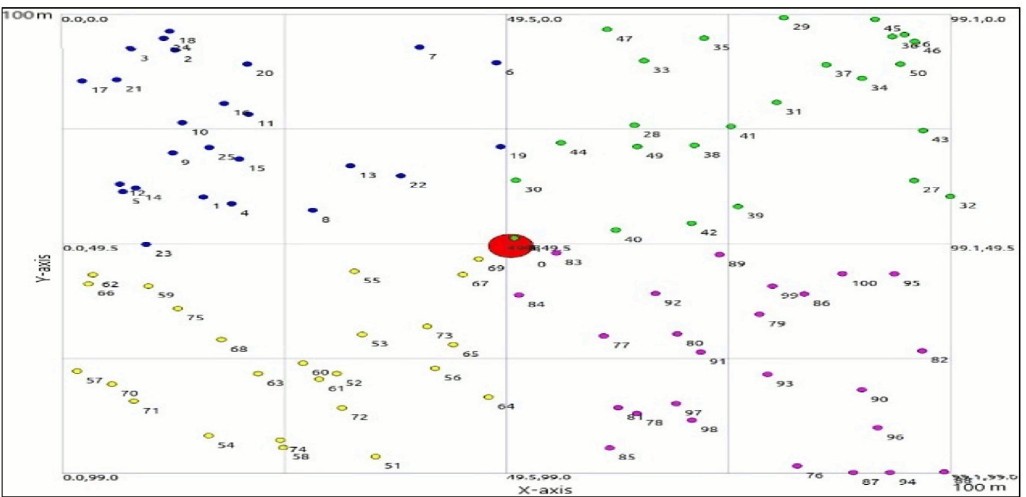

**Figure 5** Simulation environment.

its own power supply and consumes energy to operate. Energy consumption by nodes is an important aspect to consider in networking, particularly in scenarios where energy efficiency is crucial. Minimizing energy consumption can lead to cost savings, increased network reliability, and reduced environmental impact. The difference of energy consumed out of the total energy is referred to as the remaining energy or residual energy.

**Number of dead nodes**: In a routing network, the number of dead nodes refers to the count of nodes that have become non-functional or inaccessible, rendering them incapable of participating in network operations. A dead node is one that cannot transmit, receive, or process data packets, effectively becoming disconnected from the network.

$$Number\ of\ dead\ nodes = Total\ nodes - Nodes\ alive. \tag{1}$$

**Data transmission**: The quantity of data that may be effectively communicated through the network in a specific length of time is referred to as throughput. It is an indicator of how well the network can transport data. It can also be referred to as the number of packets sent and received. In the proposed model, it is the major parameter, as it determines the strength and efficacy of the proposed network. Mathematically, it is given as follows. Throughput also refers to data accuracy. More the number of packets sent, more is the data accuracy,

$$Throughput = \frac{Number\ of\ packets\ received}{Number\ of\ packets\ sent}. \tag{2}$$

## Simulation results

The section presents the results of the extensive simulations performed in the underlying work. The results are presented in the form of comparison. Overall, the results of pairing algorithm shown in Fig. 6 is appears to be part of a process for creating pairs of nodes in

**Figure 6** Pairing process.

a cluster, ensuring that each pair is unique and avoiding duplicate pairs. However, there might be a typo in the condition for skipping nodes that have already been included in pairs. This segment seems to be part of a process for pairing nodes in a cluster based on their positions and temperatures, while also considering the distance between them. Overall, the algorithm is traversing through the pairs of nodes, identifying the active nodes within each pair, and saving their IDs.

In a WSN, nodes are typically structured into clusters to professionally manage communication and data processing. Sleep-awake pairing involves managing the activity cycle of nodes within these clusters, permitting some nodes to sleep while others are active. Figure 7 likely refers to a visual representation and illustrating the clustering and pairing of nodes in the WSN. Figure 7 shows that in the 3rd cluster, there are seven pairs of nodes, and in the 4th cluster, there are four pairs. These pairs likely indicate nodes that are coordinated in their sleep-awake cycles to optimize energy usage. Within a cluster, certain nodes may have similar characteristics such as data sets and distances. In our example, nodes numbered 59, 63, 71, and 72 all have the same data set and distance. These nodes are grouped together into a pair, referred to as pair3. Pairing nodes with similar characteristics allows for more efficient coordination of sleep-awake cycles. Since these nodes have similar data sets and distances, they likely have similar communication and processing requirements, making it logical to pair them together for coordinated activity cycles. In a WSN, nodes are typically organized into clusters to efficiently manage communication and data processing. Sleep-awake pairing involves coordinating the activity cycles of nodes
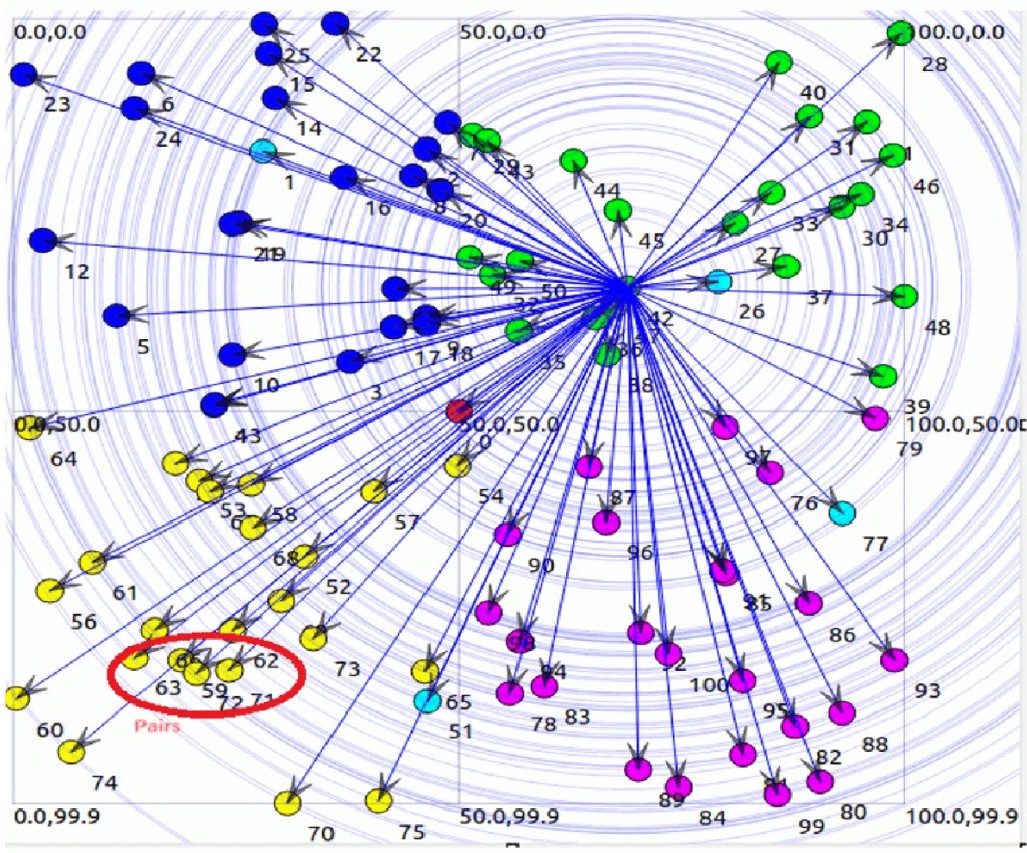

**Figure 7 Data transmission.**

within these clusters, allowing some nodes to sleep while others are active. Only node 59 is in awake state and communicates with cluster head. Figure 7 shows some aqua green nodes, these are CH and get information form awake node and send to BS.

## DISCUSSION

Overall, the algorithm of paring showing is appears to be part of a process for creating pairs of nodes in a cluster, ensuring that each pair is unique and avoiding duplicate pairs. However, there might be an error in the condition for skipping nodes that have already been included in pairs. This segment seems to be part of a process for pairing nodes in a cluster based on their positions and temperatures, while also considering the distance between them. Overall, this code segment is crossing through the pairs of nodes, identifying the active nodes within each pair, and saving their IDs.

In round 1,000, ASHNP for sure sends more packets (102,700) contrasted with EESAA (90,000) and ETASA (50,000). Additionally, in ensuing rounds (2,000, 3,000, and 4,000), ASHNP keeps on sending more packets contrasted with different procedures appearing in Fig. 8. This recommends that the ASHNP strategy reliably sends more packets across various rounds contrasted with EESAA and ETASA. It very well may be utilizing various

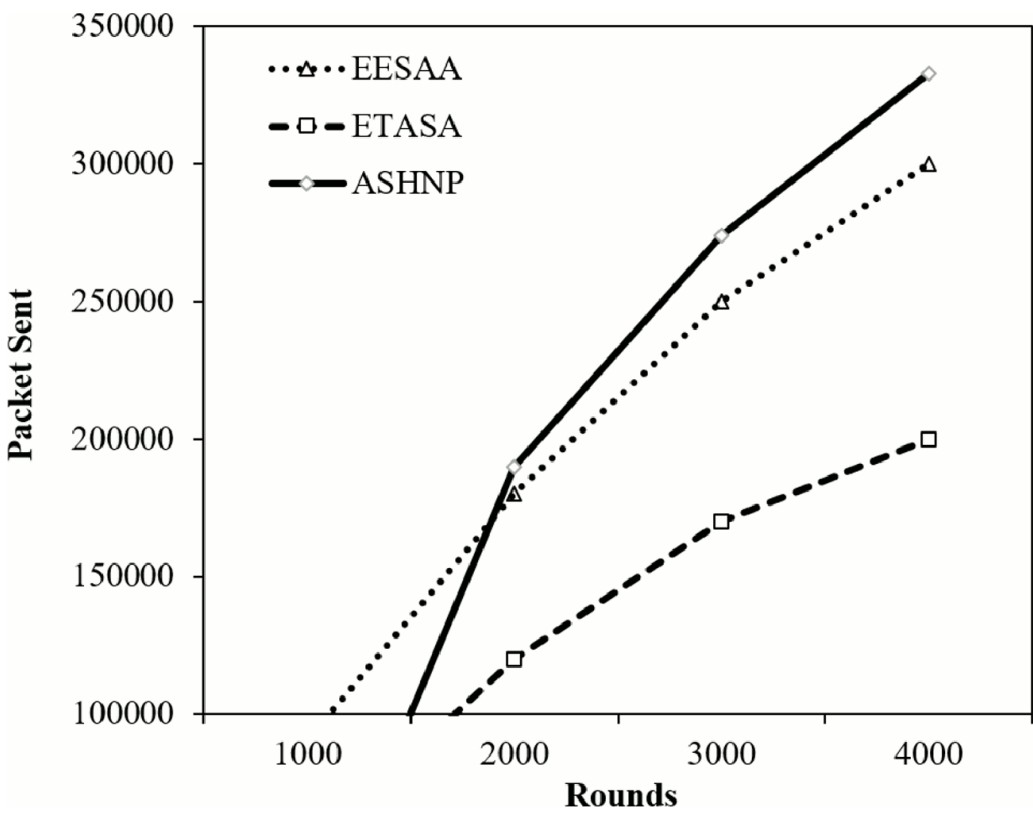

**Figure 8**  Data transmission comparison.

components or improvements bringing about higher packets transmission rates. Total number of Complete packets transmitted in round 1,000 are (when added 90,000, 50,000 and 102,700) 242,700. In round 1,000 of the WSN, the assessment of packet transmission uncovers particular examples among the strategies utilized. Among these methods, ASHNP stands apart by sending roughly 42.32% of the out all packet, a prominently higher rate contrasted with its counterparts. Interestingly, EESAA and ETASA communicated around 37.09% and 20.59% of the packets. This proposes that the ASHNP technique exhibits a more dynamic job in data transmission during this round, unbelievable different strategies as far as the volume of packets sent. Such perceptions highlight the viability and expected benefits of using the ASHNP strategy for data transmission inside the WSN clusters, possibly demonstrating its superior productivity or utilization of resources in this specific context.

Figure 9 shows the distinction of dead nodes in percentage among ASHNP and EESAA for each round, in each round, the ASHNP strategy can keep nodes alive longer by the determined rates contrasted with the EESAA. In round 1,800, the ASHNP can keep nodes alive 1.5% longer than the EESAA. As well as in round 2,500, the ASHNP can keep nodes alive 2% longer than the EESAA and in round 3,500, the ASHNP can keep nodes alive 8% longer than the EESAA. This intends that in round 4,000, the ASHNP can keep nodes alive 10% longer than the EESAA conspire. These computations show that in each round, the

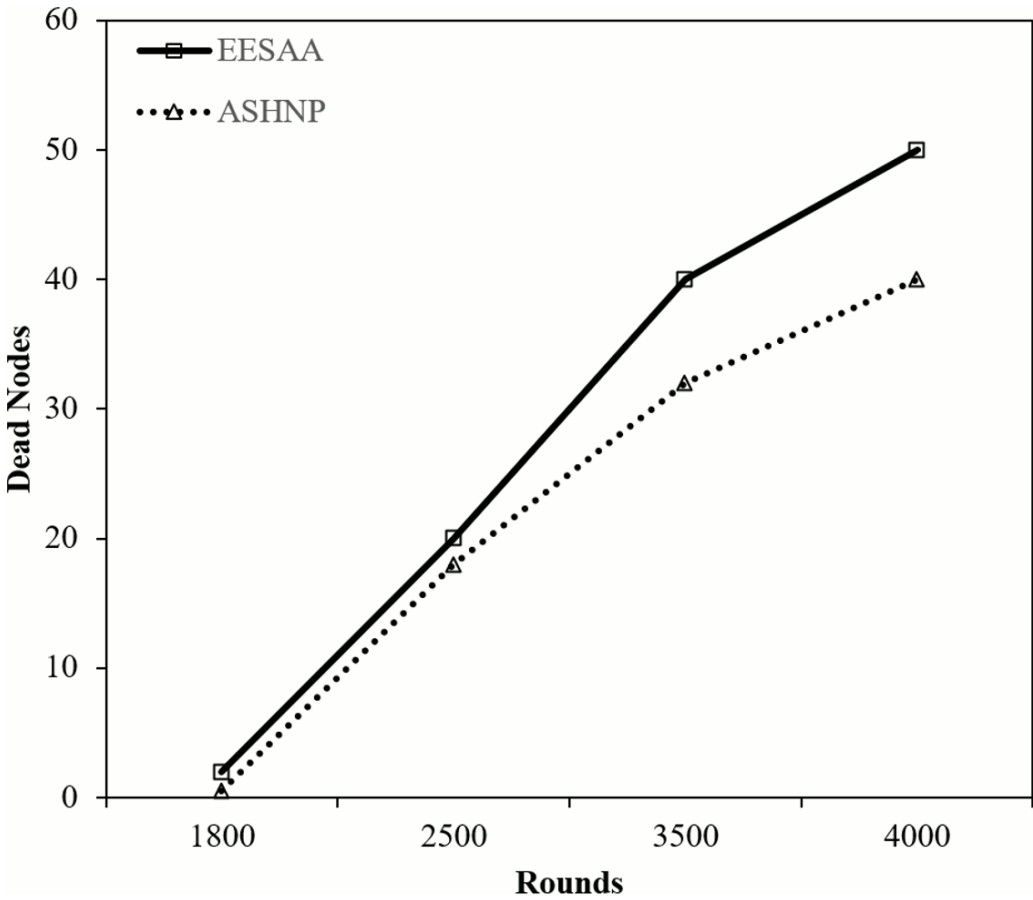

**Figure 9** Number of dead nodes.

ASHNP conspire beats the EESAA as far as keeping nodes alive, with contrasts going from 1.5% to 10%.

The energy consumption for both techniques generally decreases over the rounds, with ASHNP consistently consuming less energy compared to ETASA showing in Fig. 10. This suggests that ASHNP is more energy-efficient than ETASA across all rounds in this comparison.

The ASHNP having several key factors in-built in its design and operation. Firstly, ASHNP's innovative pairing technique ensures that nodes collaborate efficiently, leveraging their collective abilities to enhance data transmission while minimizing energy consumption. Moreover, ASHNP's adaptive approach to node pairing allows for dynamic adjustments based on real-time network conditions. Furthermore, ASHNP's success can be attributed to its full consideration of various parameters such as, distance to neighbors, and node degree in cluster head selection. By prioritizing nodes with sufficient energy reserves and preemptive positioning, ASHNP effectively allocates resources to maximize network lifetime while minimizing energy depletion. This inclusive approach ensures that ASHNP operates at high proficiency, regularly outperforming alternative techniques

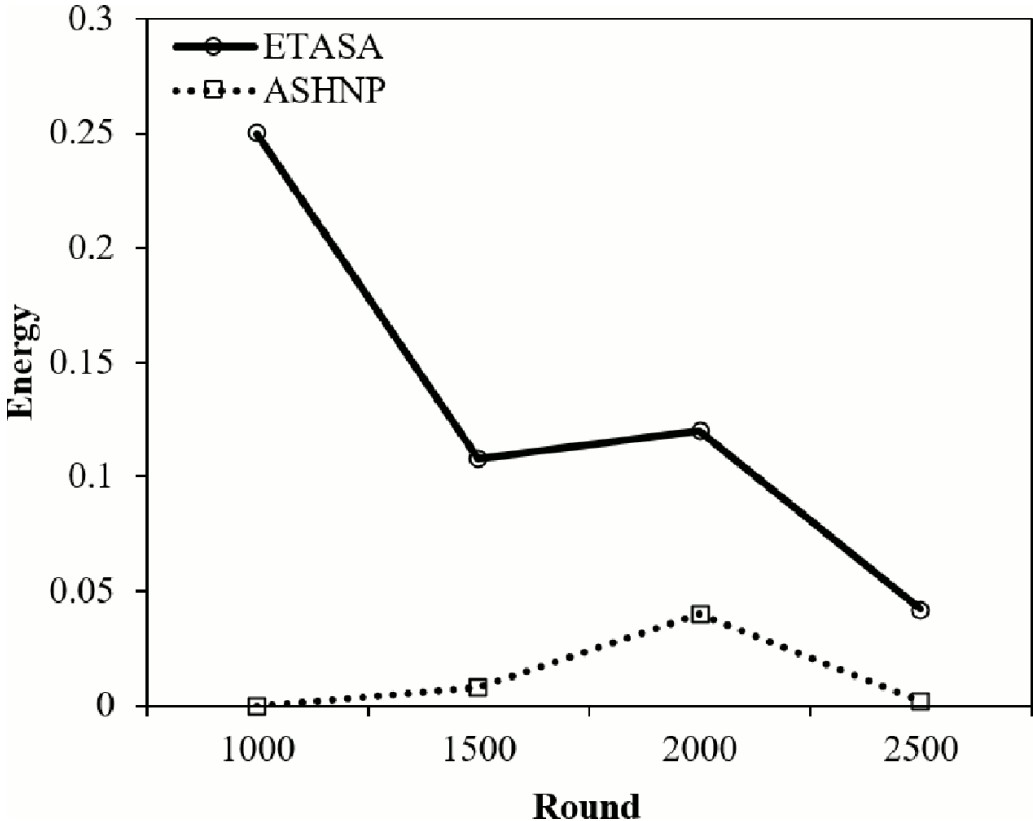

**Figure 10  Energy consumption.**

such as EESAA and ETASA. In conclusion, the tested results clearly validate ASHNP as a superior solution for WSN clustering, offering matchless performance in maintaining energy levels of nodes and certifying effective data transmission. Through its innovative design and adaptive algorithms, ASHNP sets a new standard for network optimization, setting the foundation for enhanced reliability and performance in WSN applications. In summary, the insights collected from ASHNP's results confirm its position as a primary solution for WSN clustering, offering helpful contributions to the field of wireless sensor networking and placing the foundation for future research and invention in this domain. Different acronyms used in this study have been elaborated in Table 3.

## CONCLUSION

The ASHNP algorithm's inflexibility extends to various applications, including the detection of different data rates of sensor nodes, even when they are in a sleep state. ASHNP's design, which involves paired nodes actively computing their traffic rates and jointly sensing surrounding data, offers itself well to monitoring and adapting to changes in data rates among sensor nodes. This capability enables ASHNP to detect differences in data rates, irrespective of whether the nodes are in an active or sleep state. In scenarios where sensor nodes may function at changed data rates due to environmental conditions,

**Table 3  Acronyms.**

| Acronyms | Complete word |
|---|---|
| CH | Cluster Head |
| BS | Base Station |
| WSN | Wireless Sensor Networks |
| TX | Transmission |
| SEED | Sleep Awake Energy Efficient Distributed |
| ETASA | Energy and Traffic Aware Sleep Awake |
| TEAR | Traffic and Energy Aware Routing |
| TDMA | Time Division Multiple Access |
| EESAA | Energy Efficient Sleep Awake Aware |
| ASHNP | Awake Sleep Heterogeneous Nodes' Pairing |
| SN | Sensor Nodes |

task requirements, or hardware capabilities, ASHNP's adaptive approach ensures that these differences are perfectly recognized and managed. By dynamically altering node pairing and data transmission strategies based on real-time data rate information, ASHNP optimizes network efficiency and responsiveness, thereby enhancing overall performance. The utilization of ASHNP for detecting different data rates of sensor nodes, even during periods of dormancy, illustrates its efficiency in addressing the miscellaneous challenges faced in WSNs. As WSN applications continue to grow, ASHNP's adaptability positions it as a valued tool for maximizing data transmission efficiency and facilitating intelligent decision-making across various domains and industries.

### Funding
No funding was received to complete this research work.

### Competing Interests
The authors declare there are no competing interests.

### Author Contributions
- Zahida Shaheen conceived and designed the experiments, performed the experiments, analyzed the data, performed the computation work, prepared figures and/or tables, authored or reviewed drafts of the article, and approved the final draft.
- Kashif Sattar conceived and designed the experiments, performed the experiments, analyzed the data, performed the computation work, authored or reviewed drafts of the article, and approved the final draft.
- Mukhtar Ahmed analyzed the data, prepared figures and/or tables, authored or reviewed drafts of the article, and approved the final draft.

### Data Availability
The raw data and code are available in the Supplemental Files.

## Supplemental Information

Supplemental information for this article can be found online at http://dx.doi.org/10.7717/peerj-cs.2243#supplemental-information.

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
