# Peer review of "Pairing algorithm for varying data in cluster based heterogeneous wireless sensor networks"

_PeerJ Computer Science, doi:10.7717/peerj-cs.2243_

## Round 0.1 · original submission · Major Revisions

Some comments should be considered as below:
1. In the "Abstract", please provide the quantitative performance achievement of the algorithm.
2. In the "Introduction", the problem statement should be clearly defined.
3. The literature review should be revised to clearly indicate the knowledge gap that authors would like to address in the work.
4. In the simulated results, the comparison of the proposed method with other works should be considered. Verification of the results should be enhanced as well if possible.

Reviewer 2 has requested that you cite specific references. You may add them if you believe they are especially relevant. However, I do not expect you to include these citations, and if you do not include them, this will not influence my decision.

**Language Note:** The review process has identified that the English language must be improved. PeerJ can provide language editing services - please contact us at [email protected] for pricing (be sure to provide your manuscript number and title). Alternatively, you should make your own arrangements to improve the language quality and provide details in your response letter. – PeerJ Staff

Reviewer 1 ·

Basic reporting

The manuscript entitled “Pairing algorithm for varying data in cluster based heterogeneous Wireless Sensor Networks” has been investigated in detail. The manuscript presents a new routing algorithm, the Awake Sleep Heterogeneous Nodes Pairing (ASHNP) scheme, for Wireless Sensor Networks (WSNs). Addressing energy efficiency and load balancing, ASHNP employs a pairing strategy based on data uniformity within node vicinity. The manuscript highlights simulation results showing improvements in metrics like dead nodes and throughput. While comparing ASHNP with Energy Efficient Sleep Awake Aware (EESAA), the abstract emphasizes its feasibility. However, the manuscript lacks clarity in describing ASHNP's core principles and its novelty compared to existing methods. More detailed explanations and clearer presentation of simulation results would enhance its effectiveness. There are some points that need further clarification and improvement:
1) The introduction sets the context for the need for efficient routing algorithms in wireless sensor networks (WSNs) but lacks a clear statement of the problem and its significance. It could benefit from a more concise and focused introduction that highlights the specific challenges addressed by the proposed algorithm.
2) The literature review briefly mentions existing methods like SEED, TEAR, and ETASA but fails to provide a comprehensive overview of related work in the field. A more detailed comparison and analysis of existing routing algorithms would strengthen the paper's contribution.

Experimental design

The simulation results are mentioned but not discussed in detail. The paper should provide a thorough analysis of the simulation findings, including comparisons with existing algorithms like EESAA. Without a detailed discussion of the results, it's challenging to evaluate the effectiveness of the proposed ASHNP scheme.

The use of technical terminology like "pairing strategy" and "heterogeneous sensor nodes" is appropriate but could be explained more clearly for readers unfamiliar with WSNs. Simplifying complex terms and providing clear definitions would improve the paper's accessibility and readability.

Validity of the findings

“Simulation Results’ Discussion” section should be edited in a more highlighting, argumentative way. The author should analysis the reason why the tested results is achieved.

Additional comments

The manuscript entitled “Pairing algorithm for varying data in cluster based heterogeneous Wireless Sensor Networks” has been investigated in detail. The manuscript presents a new routing algorithm, the Awake Sleep Heterogeneous Nodes Pairing (ASHNP) scheme, for Wireless Sensor Networks (WSNs). Addressing energy efficiency and load balancing, ASHNP employs a pairing strategy based on data uniformity within node vicinity. The manuscript highlights simulation results showing improvements in metrics like dead nodes and throughput. While comparing ASHNP with Energy Efficient Sleep Awake Aware (EESAA), the abstract emphasizes its feasibility. However, the manuscript lacks clarity in describing ASHNP's core principles and its novelty compared to existing methods. More detailed explanations and clearer presentation of simulation results would enhance its effectiveness. There are some points that need further clarification and improvement:
1) The introduction sets the context for the need for efficient routing algorithms in wireless sensor networks (WSNs) but lacks a clear statement of the problem and its significance. It could benefit from a more concise and focused introduction that highlights the specific challenges addressed by the proposed algorithm.
2) The literature review briefly mentions existing methods like SEED, TEAR, and ETASA but fails to provide a comprehensive overview of related work in the field. A more detailed comparison and analysis of existing routing algorithms would strengthen the paper's contribution.
3) The description of the Awake Sleep Heterogeneous Nodes Pairing (ASHNP) scheme is somewhat clear but lacks depth in explaining the underlying principles and mechanisms. The paper could benefit from a more detailed explanation of how ASHNP addresses the identified drawbacks of existing methods.
4) The simulation results are mentioned but not discussed in detail. The paper should provide a thorough analysis of the simulation findings, including comparisons with existing algorithms like EESAA. Without a detailed discussion of the results, it's challenging to evaluate the effectiveness of the proposed ASHNP scheme.
5) The use of technical terminology like "pairing strategy" and "heterogeneous sensor nodes" is appropriate but could be explained more clearly for readers unfamiliar with WSNs. Simplifying complex terms and providing clear definitions would improve the paper's accessibility and readability.
6) “Simulation Results’ Discussion” section should be edited in a more highlighting, argumentative way. The author should analysis the reason why the tested results is achieved.
7) It will be helpful to the readers if some discussions about insight of the main results are added as Remarks.
Overall, the paper presents a novel routing algorithm, ASHNP, for wireless sensor networks. While the idea of addressing energy efficiency and load balancing is commendable, the paper lacks depth in several areas, including the literature review, method description, and simulation analysis. To enhance its contribution, the paper should provide a clearer problem statement, a more comprehensive literature review, detailed explanation of the proposed method, thorough simulation results, and analysis. With these improvements, the paper could make a stronger contribution to the field of WSN routing algorithms. This study may be proposed for publication if it is addressed in the specified problems.

Reviewer 2 ·

Basic reporting

Please see comments below

Experimental design

Please see comments below

Validity of the findings

Please see comments below

Additional comments

1) Please collect all the acronyms employed throughout the paper in a given table for readers’ convenience.

2) The related work discussion should be complemented with a table categorizing different algorithms/approaches along their main distinctive characteristics so as to better position the present proposal.


3) In general, the proposed routing algorithm should be qualitatively discussed to the usual inference problems which are tackled in clustered WSNs, e.g.:
Aldalahmeh, Sami A., and Domenico Ciuonzo. "Distributed detection fusion in clustered sensor networks over multiple access fading channels." IEEE Transactions on Signal and Information Processing over Networks 8 (2022): 317-329.


Tian, Qingjiang, and Edward J. Coyle. "Optimal distributed detection in clustered wireless sensor networks." IEEE Transactions on Signal Processing 55.7 (2007): 3892-3904.


Wu, Linlong, et al. "Optimization Based Sensor Placement for Multi-Target Localization With Coupling Sensor Clusters." IEEE Transactions on Signal and Information Processing over Networks (2023).

4) I recommend that the authors add a system model figure to complement the discussion of the system model section so as to provide a bird’s eye view of the considered WSN setup.

5) The statement of contributions should be enriched/detailed by highlighting the main technical challenges tackled by the authors in the context of WSN energy-efficiency.


6) Please add a paper organization paragraph at the end of the introduction section.

7) I have found the system flowchart depicted in Fig. 2 very hard to follow and needing some improvement to provide a more structured exposition.


8) In my opinion, more details should be provided on sensing capabilities of the sensor devices considered in this study and the corresponding numerical analysis.

9) The authors should relate the simulation setup considered to a practical application scenario in WSNs to better motivate the considered set of parameters.


10) Conclusion section should be enriched with a brief paragraph describing future research directions.

Reviewer 3 ·

Basic reporting

The paper titled “Pairing algorithm for varying data in cluster based heterogeneous Wireless Sensor Networks”. This author claimed to propose an efficient routing algorithm termed as Awake Sleep Heterogeneous Nodes Pairing (ASHNP) scheme. The proposed approach utilized the unique pairing strategy to decrease data loss. The simulation results prove the proposed works efficacy in terms of various metrics like number of dead nodes, throughput, and number of packets sent.
The paper is not clearly written in a good style and lack of cohesiveness in introduction and other parts. Overall I don’t recognize any novelty and also literature study is very basic and insufficient.

Experimental design

The proposed approach is very basic and also experimental set up is very basic.

.

Validity of the findings

Not sufficient details provided.

Additional comments

As per my review paper is not as per minimum standard set by journal. So my advice is Rejection.

---

## Round 0.2 · accepted · Accept

Authors have addressed all the reviewers' comments satisfactory. Hence, this paper is recommended for Acceptance in its current form.

Reviewer 1 ·

Basic reporting

My comments have been addressed. It is acceptable in the present form.

Experimental design

My comments have been addressed. It is acceptable in the present form.

Validity of the findings

My comments have been addressed. It is acceptable in the present form.

Reviewer 2 ·

Basic reporting

The authors have addressed my previous comments.

Experimental design

The authors have addressed my previous comments.

Validity of the findings

The authors have addressed my previous comments.

Additional comments

The authors have addressed my previous comments.